behaviour/computational biology/ecology

coordination, decision-making, heuristics, rules-of-thumb, self-organization, collective behaviour

**Authors for correspondence:**
D. W. E. Sankey
e-mail: d.sankey@exeter.ac.uk
A. J. King
e-mail: a.j.king@swansea.ac.uk

†Present address: Centre for Ecology and Conservation, University of Exeter, Penryn Campus, Penryn, Cornwall, TR10 9FE, UK.

# Consensus of travel direction is achieved by simple copying, not voting, in free-ranging goats

D. W. E. Sankey[1,†], L. R. O'Bryan[2], S. Garnier[2], G. Cowlishaw[3], P. Hopkins[1], M. Holton[1], I. Fürtbauer[1] and A. J. King[1]

[1]Swansea University, Singleton Park, Swansea SA2 8PP, UK
[2]New Jersey Institute of Technology, Newark, NJ 07102, USA
[3]Institute of Zoology, Zoological Society of London, Regent's Park, London NW14RY, UK

DWES, 0000-0002-6363-8023; SG, 0000-0002-3886-3974; IF, 0000-0003-1404-6280; AJK, 0000-0002-6870-9767

For group-living animals to remain cohesive they must agree on where to travel. Theoretical models predict shared group decisions should be favoured, and a number of empirical examples support this. However, the behavioural mechanisms that underpin shared decision-making are not fully understood. Groups may achieve consensus of direction by active communication of individual preferences (i.e. voting), or by responding to each other's orientation and movement (i.e. copying). For example, African buffalo (*Syncerus caffer*) are reported to use body orientation to vote and indicate their preferred direction to achieve a consensus on travel direction, while golden shiners (*Notemigonus crysoleucas*) achieve consensus of direction by responding to the movement cues of their neighbours. Here, we present a conceptual model (supported by agent-based simulations) that allows us to distinguish patterns of motion that represent voting or copying. We test our model predictions using high-resolution GPS and magnetometer data collected from a herd of free-ranging goats (*Capra aegagrus hircus*) in the Namib Desert, Namibia. We find that decisions concerning travel direction were more consistent with individuals copying one another's motion and find no evidence to support the use of voting with body orientation. Our findings highlight the role of simple behavioural rules for collective decision-making by animal groups.

# 1. Introduction

To reap the benefits of group-living, animals must make collective decisions regarding the timing and direction of movement [1–3]. Theoretical models predict that shared decision-making—where a majority of group members contribute to decisions—produces less extreme outcomes [2,4] and can result in greater decision accuracy [5–7] compared with unshared decisions dominated by an individual or minority of group members. While shared decision-making occurs in many species [8–12], the behavioural mechanisms underlying such decisions are not fully understood.

One proposed behavioural mechanism for shared group decision-making is that members communicate individual preferences (i.e. 'vote') with regard to the decision outcome before the onset of a collective movement [4,8,13–15]. Many examples from the literature involve vocalizations. For example, meerkats (*Suricata suricatta*) and African wild dogs (*Lycaon pictus*) emit vocalizations prior to collective movements and only when a sufficient number of individuals vocalize does the group depart [16,17]. However, animals are also reported to vote with their body orientations; the most famous example of this being African buffalo (*Syncerus caffer*). Prins [14] described how, during rest periods, African buffalo use their body orientation to indicate their preferred direction of travel, and when the herd begins to move after resting, the group moves in the average direction indicated by group members' orientations. Hamadryas baboons (*Papio hamadryas*) and Tonkean macaques (*Macaca tonkeana*) are reported to use a similar voting mechanism—with individual body orientation indicating individuals' preferred movement direction [11,18]. These studies are consistent with the hypothesis that communication enables a type of structured decision-making whereby each group member is able to assess the relative support for different options among their group-mates [4].

In contrast to structured voting mechanisms, shared group decisions concerning travel initiation and direction can emerge from relatively simple local interaction rules [19]. In fact, theoretical models and empirical data have demonstrated that individuals that do not have relevant prior information concerning the decision at hand can play an important role in consensus decision-making; enforcing equal representation of preferences and promoting a democratic outcome [19]. Under this scenario, individuals copying each other's movement (e.g. attraction to each other's positions and/or orientations) during a collective movement results in a consensus across individuals' directional preferences in both real and simulated animal groups [20,21]. Empirical evidence in support of copying mechanisms mostly comes from short-term observations of shoaling fish in laboratory settings [9,22,23]. Moreover, analyses of the movement trajectories for a troop of wild olive baboons (*Papio anubis*) suggests that simple copying and averaging of directional preferences may underlie group decisions about where to travel in wild, socially complex animal societies [12].

While there is evidence for both voting and copying mechanisms in animal systems, differentiating between these two mechanisms is challenging, not least because they can produce similar individual and group-level behaviours [24,25]. Indeed, distinguishing between voting and copying mechanisms in the wild requires that researchers continuously and simultaneously measure group members' orientation and movement towards a target destination, which is logistically challenging [12,26–28]. Here, we use a conceptual model, agent-based simulations, and empirical data from a herd of free-ranging goats (*Capra aegagrus hircus*) at the edge of the Namib Desert, to distinguish patterns of motion that represent voting or copying.

Goats are highly gregarious and make collective decisions about the nature and timing of their activities [29,30]. Preliminary observations of goat behaviour at the study site indicated the goats move and forage through a highly patchy (dry) riverbed, without observable consistencies in foraging patch choice. When transitioning from resting to moving in this habitat, the goats showed high cohesiveness without any identifiable leaders or obvious signalling, which is consistent with a shared decision-making process and previous observations of other social ungulates' collective movements (e.g. *Bos indicus*: [31]; *Bos taurus*: [32]). We, therefore, tested the hypotheses that goats either: (i) used body orientation to vote on their preferred direction of travel *prior to departure* (voting hypothesis) [11,14,18] or (ii) simultaneously matched their orientation and trajectory with neighbours *as the group moves off* (copying hypothesis) [25,33,34].

To differentiate between our hypotheses, we fitted all group members with bespoke GPS- and inertial sensor-enabled collars that allowed us to determine the heading and position of individuals every second. Using these data, we define a 'decision parameter'—the congruence of group mean orientation and departure direction—at every time step before identified collective movements, with values ranging between 0 (opposite angles) and 1 (perfect agreement of angles). The voting hypothesis predicts the decision parameter will increase *before* the group speeds up (figure 1*a*,*b*), i.e. they will 'point, then move'. The maximum correlation between decision parameter and speed should therefore have a negative time

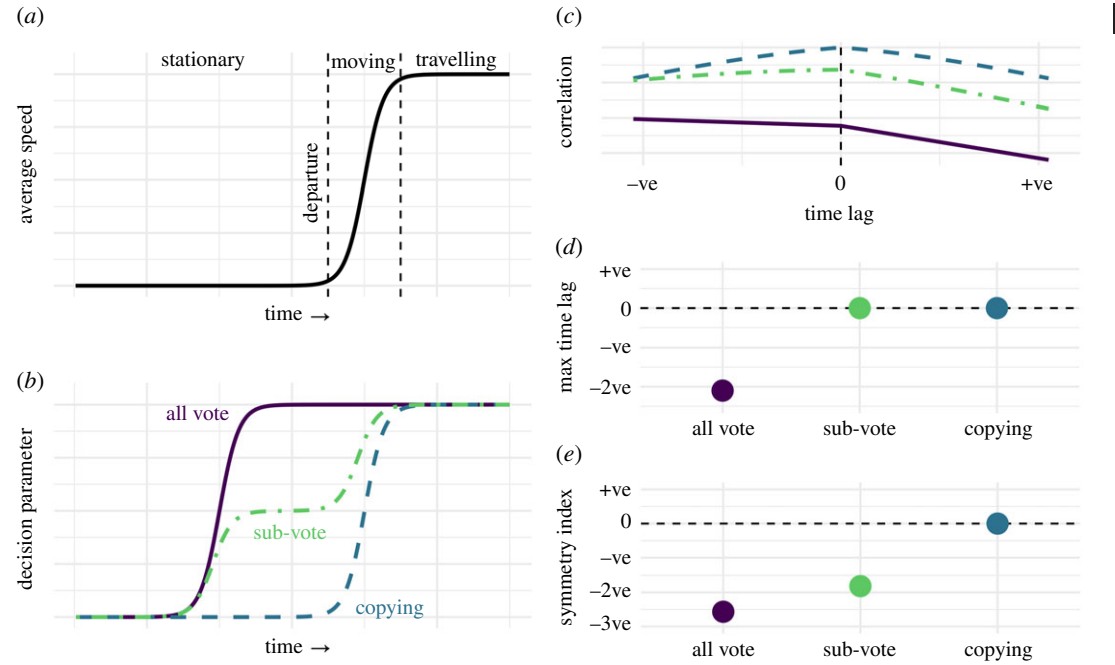

**Figure 1.** Graphical representation of hypotheses and predictions. (*a*) Group mean speed as a function of time, during a collective decision. First, individuals are slow- or non-moving, labelled as the 'stationary' phase. Then, individuals begin to depart, and group mean speed increases, labelled as the 'moving' phase. Finally, the group are travelling together towards a new destination, labelled as the 'travelling' phase. (*b*) Group alignment towards their next destination 'decision parameter' as a function of time, during a collective decision. Predictions for 'all vote', 'sub-vote' and 'copying' mechanisms are shown and the timeline matches that shown in (*a*). The decision-parameter quantifies when the group members' mean heading is pointed towards (high values) or away from (low values) their next destination. If individuals vote on their preferred direction, the decision-parameter should show a sigmoid-like response and be maximized before the group departs during the stationary phase (purple solid line). If a sub-group votes on their preferred direction, then decision-parameter should show a two-step increase; the first occurs during the stationary phase before departure and the second increase during the departure phase, when the remaining group members follow the voting individuals' lead (green dotted and dashed line). If no voting occurs, but instead copying behaviour is employed, individuals should respond to the motion/heading of their neighbours, resulting in sigmoid-like response at approximately the same time as the individuals begin to depart (blue dashed line). (*c*), (*d*) and (*e*) show the expected statistics when calculating the cross-correlation for decision-parameter and speed under the different scenarios. *Max time lag* is the lag in time for which the cross-correlation value is maximal. For 'all vote' we expect maximum cross-correlation to be negative lag (i.e. decision-parameter changes preceded speed changes), and close to zero for 'sub-vote' and 'copying' (because decision-parameter is highest as the group departs). *Symmetry index* represents the imbalance between the correlation values on the left (negative time lags) and right (positive time lags) sides of the cross-correlation output. The area under the curve to the left of zero is subtracted from the area under the curve to the right of zero. Negative time lags indicate that decision-parameter changes preceded speed changes, which should be greatest for 'all vote', and intermediate for 'sub-vote'. 'Copying' symmetry index should be close to zero.

lag if voting is used (figure 1*c,d*). The copying hypothesis predicts that individuals respond to the motion/heading of their neighbours (e.g. via simple attraction), resulting in a consensus of direction at the *same time* as departure (figure 1*a,b*). The maximum correlation between speed and the decision parameter should therefore be almost instantaneous if copying is used (figure 1*c,d*). Figure 1 and electronic supplementary material, video S1 provide graphical representations of our hypotheses and predictions. To aid interpretation of our findings we created agent based models for 'voting' and 'copying' and compare these model outputs with our conceptual model (figure 1) and empirical data. Finally, we also investigated pairwise time-lag correlation analyses of goat headings during the departure phase and when travelling to explore consistency in goats' responses to one another's change in headings [35,36].

# 2. Material and methods

## 2.1. Study subjects and data collection

Fieldwork was carried out at Tsaobis Nature Park, Namibia (22°23′ S, 15°45′ W), with a group of adult female goats (*n* = 16). Goats were kept in the corral overnight but were free to roam for 5–6 h per day

from early morning to early afternoon. See O'Bryan *et al.* [30] for further information on housing and husbandry. Goats were fitted with modified SHOALgroup F2HKv2 collars (see [37] for details). Each collar contained a 'Technosmart GiPSy 4' GPS, recording geolocation at 1 Hz, and a tri-axes accelerometer/magnetometer 'Daily Diary' sensor [38] recording motion data at 40 Hz. The goats wore the collars for a 10-day period (9–18 September 2015) and we present analyses of data from $n = 10$ goats for which we have complete GPS and motion data over this period. All data retrieved were calibrated to ensure correct orientation with respect to the animal (electronic supplementary material S1; figures S1–S3), before undertaking analyses described below.

## 2.2. Heading and speed

For each time step ($t = 1$ s), we recorded each individual's orientation with respect to north, the location of the group's centroid (the mean latitude and longitude of all individuals' GPS locations), and the group's mean orientation (the circular mean of all individuals' orientations from magnetometers—R package: CircStats [39]). Using the next time step ($t + \Delta t$), we calculated individuals' instantaneous speed and heading, between successive GPS locations. We used magnetometer-derived orientation and confirmed the accuracy of these data by examining the correlation between magnetometer-derived orientation and GPS heading when goats were fast-moving (greater than 1.25 m in the same direction over 5 s; percentile = 0.78). There was a significant correlation between the magnetometer heading and GPS heading over these stretches (Pearson's product moment coefficient = 0.88 $n = 7001$, $p < 0.001$; see electronic supplementary material, S2 and figure S4 for more details).

## 2.3. Group decisions using change-point analyses

We identified potential collective decisions from resting to moving by using change-point analyses [40] to detect statistically significant changes in group heading and/or speed ($n = 73$) (electronic supplementary material, S3, figures S5, S6). We then discarded any decisions for which we could not also identify a clear pre-departure (stationary) period, and instances where movement could have been influenced by researchers (namely, the first and last decisions of the day), leaving $n = 23$ decisions (electronic supplementary material, S4). We identified the departure direction as the strongest change-point in the direction (heading) and directedness (heading variance) following a pre-departure window (electronic supplementary material, S4, figures S7, S8). Decision parameter (see below and figure 2) by default rises to 1 at this point (electronic supplementary material, figure S9), allowing us to test whether speed was concurrent with this rise or not. No change-points were identified in two of our pre-departure windows, providing us with $n = 21$ decisions for testing our hypothesis and predictions. To check the validity of the criterion we used in our change-point analyses (electronic supplementary material, S3 and S4), we performed sensitivity analyses and found that all results were qualitatively similar and interpretations unchanged (electronic supplementary material, S5).

## 2.4. Decision parameter

For each time step, ranging from 100 s before to 100 s after each identified collective decision, we calculated a decision parameter ($r$) (figure 2), using the length of the summed vector for group mean orientation and departure direction at each time step (1 Hz), where $\alpha$ is (1) group mean heading and (2) departure direction for $i = 1$ and $i = 2$, respectively (figure 1; electronic supplementary material, video S1), calculated as follows:

$$r = \sqrt{\left( \left( \frac{\sum_{i=1}^{2} \cos_{\alpha(i)}}{2} \right)^2 + \left( \frac{\sum_{i=1}^{2} \sin_{\alpha(i)}}{2} \right)^2 \right)}. \tag{2.1}$$

To estimate whether decision parameter increased before, at the same time as, or after speed, we cross-correlated decision parameter with group mean speed (base R: [41]; figure 1c). We recorded the time lag where the cross-correlation curve was maximal (CorrMax; figure 1d) and the symmetry of the cross-correlation curve with respect to zero (Sym; figure 1e). These statistics were re-sampled with replacement ($n = 1000$ bootstrap samples) producing confidence intervals for the median value. Using mean average made no qualitative difference to the direction or significance of the statistics, but median was a better fit to the non-normal data.

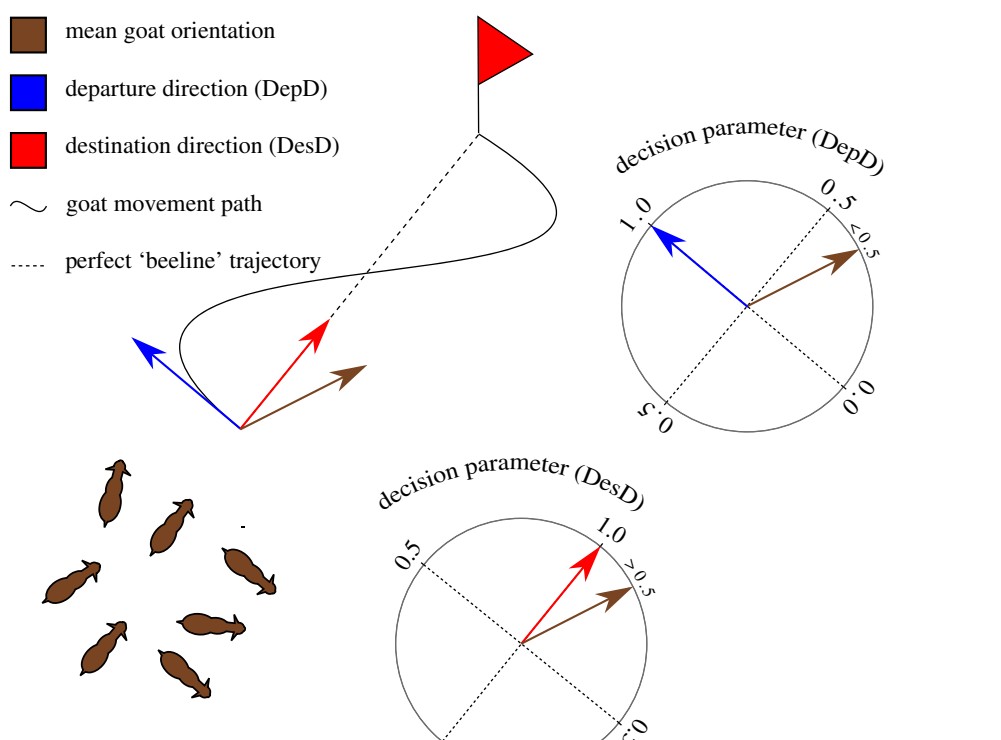

**Figure 2.** Decision parameter schematic. At each time step before, during and after a collective departure (using 1 Hz animal-attached GPS), we used decision parameter ($r$) (bottom right) to estimate how closely the group's mean orientation (brown arrow) matched the departure direction (blue arrow) or destination direction (red arrow). Decision parameter must rise to 1 as a group departs, allowing us to test whether such an increase is concurrent with an increase in speed (copying hypothesis) or precedes it (voting hypotheses).

## 2.5. Departure direction or destination direction?

The decision parameter estimates how closely the group's orientation matched the departure direction at every time step before, during and after departure. We focused on departure direction because this needs to be realized—and rise to a value of 1—at some point during a collective departure window (see above and figure 1). Our question was whether departure direction is realized *at the same time as*, or *before* the speed increase. Destination direction, on the other hand, does not necessarily need to be reached in collective departure windows. While destination direction is still a useful metric, it was not the primary focus of this study. Nevertheless, we explicitly tested destination direction using the same methods we applied to departure direction. We found no statistical difference (see Results), probably due to the fact that departure direction and destination direction were strongly correlated in our study (Pearson's correlation coefficient; $r = 0.821$, $n = 21$, $t = 3.737$, $p < 0.001$).

## 2.6. Polar order parameter

To check goat alignment regardless of the group mean orientation and decision parameter, we also calculated a polar order (or, alignment) parameter using the 'rho.circular' function (R package: CircStats; [39]). This parameter is required to confirm group members aligned with each other during departure (and not only to the travel destination). At each time step, we converted the goats' headings into unit vectors, which were then averaged to calculate the polar order parameter (0/1 = all goats pointing in equally opposite/the same direction).

## 2.7. Goat copying

To explore whether goats showed consistency in their response to one another's change in headings we performed pairwise time-lag correlation analyses of goat heading data (from magnetometers) 100 s before and after departures and assigned individuals in dyads a value of −1 (following), +1 (leading) or 0 (neutral) depending on whether the maximum correlation of heading was positive, negative, or

at no shift in time. We created a matrix and populated cells with the summed scores for all pairs of goats, ranking pairs by score to allow us to test for hierarchical structure in this network with R package 'compete' [42,43]. Further details of how we determined changes in heading and relationships between dyad headings are given in electronic supplementary material, S6.

## 2.8. Agent-based model

To aid interpretation of our findings, we created agent-based models (*post hoc*) for comparison with our conceptual model and empirical data. More details on the implementation of the model are provided in electronic supplementary material, S7. Briefly, at the start of each model simulation, agents were cohesive (locations normally distributed in $x$ and $y$ directions) and stationary (with speed = 0 units per second). Then, following a 'movement-initiation' time step (sampled from a normal distribution), the agents start to move (at $\sigma$ units per second). This produces an increase in group speed as a sigmoid curve after movement initiation (electronic supplementary material, figure S10) as outlined in our conceptual model (figure 1a). When moving, the agents also turn toward the circular mean of their neighbour orientations. Agents turn with a maximum turn rate of 0.75 radians per time step, taken from empirical measurements (only 5% of turning angles were greater than 0.75 rad s$^{-1}$) and all neighbours were included in the circular mean calculation, as goats did not align more sharply with their nearest neighbour than any other given neighbour (from second to ninth) (ANOVA: d.f.$_{1,187108}$, $F = 2.047$, $p = 0.153$). In fact, there was a slight but not significant increase in the turning rate towards further neighbours, (LM: d.f. = 187108, $t = 1.429$, $p = 0.153$).

For the 'copying' version of the model, simulations were run as above. For the 'voting' version of the model, we added a 'sub-vote', or 'all vote' condition in the stationary phase, where agents orient toward an angle drawn from a von Mises distribution (electronic supplementary material, figure S11), and we termed this the 'vote initiation'. Like the movement initiation, vote initiation time was sampled from a normal distribution.

For each iteration run, we recorded the group mean speed, decision parameter and polar order parameter for each time step as was taken from the empirical data. We subsampled 21 decisions from the model 1000 times with replacement to produce datasets the same size as our empirical dataset. For each of these 1000 subsamples, we treated the data as in the empirical dataset, estimating (i) cross-correlation maximum, and (ii) symmetry, using 1000 bootstrap permutations. Finally, we corroborated all of these estimates to provide a mean and a standard deviation of these prediction statistics (figure 3; also see electronic supplementary material, S7). Topological structure (taking a circular mean of a focal individual's closest $n$ neighbours) was also added to models (i.e. not for the main predictions) to investigate the effect of topological structure on cohesion (electronic supplementary material, figure S12).

# 3. Results

Prior to the group decision to move towards a new location, goats were slow-moving or stationary (figure 4a). Then, at departure, mean speed (figure 4a) and decision parameter (figure 4b) increase in a sigmoid-like response (matching predictions: figure 1a,b). Polar order parameter also increased, indicating that group members aligned with each other during departure (figure 4c). Following departure, groups maintained their average speed (figure 4a) and travelled towards their next destination (figure 4b).

One hundred seconds prior to the identified collective movement, the decision parameter was already higher than random, with decision parameter values around 0.76 (mean value) (one sample $t$-test; d.f. = 20, $t = 2.45$, $p = 0.02$; figure 4b). This is reflected in that most decisions were biased in the direction of the previous movement. The decision parameter at termination of previous movement was 0.68 (mean value), and there was no increase in decision parameter from termination values and the mean value at 100 s before the departure (one sample $t$-test; d.f. = 20, $t = -0.87$, $p = 0.39$).

During the movement phase (figure 4a) the group became more aligned with the departure direction, identified by a steep rise in decision parameter. The maximum cross-correlation between decision parameter and speed was close to 0 s (mean [95% confidence intervals] = $-1.1$ [$-5.1$; 3.0]; figure 4d, 3e), supporting predictions of both sub-vote and copying hypotheses from the conceptual model (figure 1d). The symmetry of the cross-correlation was also close to zero (mean [95% confidence intervals] = 0.1 [$-1.5$; 1.5]; figure 4e), which means that the alignment towards the departure direction increased at the same time as travel speed. Our results therefore support the predictions of the copying hypothesis (figure 1).

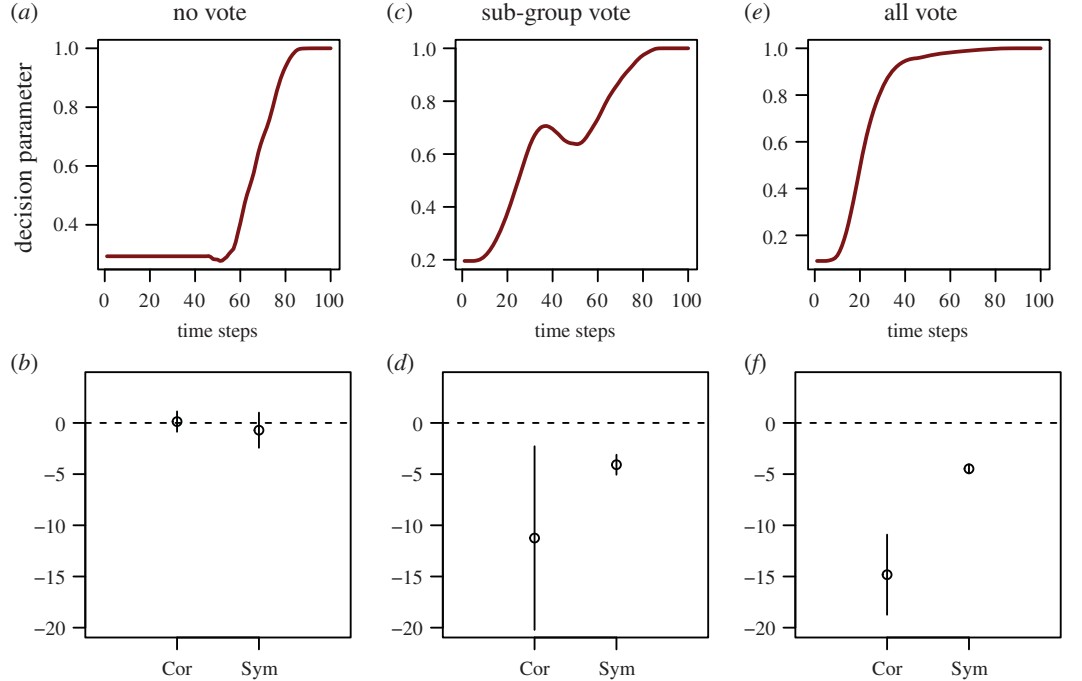

**Figure 3.** Agent-based model predictions with 'copying' (i.e. no vote) (*a,b*), 'sub-group vote' (*c,d*) or 'all vote' (*e,f*) mechanism. Plots show decision parameter curve (top row) and prediction statistics (mean and 95% confidence intervals) for (i) the time at which maximum correlation is seen (Cor), and (ii) symmetry of the cross-correlation curve (Sym) for the decision parameter crossed with group speed. All prediction statistics confidence intervals overlap predictions from our conceptual model, except maximum correlation point for the sub-group vote curve (in *d*), which is highly variable and has a negative predicted value rather than zero. Statistics were generated using 1000 permutations of $n = 21$ collective decisions, following the same method as was used for the empirical dataset.

Our agent-based model predictions generally matched those of our conceptual model (figure 3; electronic supplementary material, S7). For 'copying' and 'all-vote' scenarios the model outputs were identical to our conceptual approach, while for the 'sub-group vote' scenario, the maximum of the cross-correlation curve was predicted to be negative (same as predicted for the 'all vote' scenario) (figure 3). Comparing our empirical data and the agent-based models (figure 3*a,b*) therefore further support the copying hypothesis.

Time-lag correlation analyses of goat headings showed little transitivity prior to departure (−60 s to 0 s as indicated in figure 4: −0.13, $p = 0.729$) or during the travelling phase (0 s–60 s as indicated in figure 4: −0.067, $p = 0.618$), but significant directional consistency (prior to departure: 0.185, $p = 0.016$; travelling phase: 0.202, $p < 0.001$). This indicates preferential copying of goat headings (e.g. A is more likely to copy the orientation of B than B is to copy A), but does not indicate structured (hierarchical) copying across the group (i.e. no transitivity; A follows B and B follows C does not mean that A necessarily follows C).

## 4. Discussion

To investigate mechanisms underlying collective travel decisions, we measured animal orientation and motion at high resolution in free-ranging conditions. Our results do not support a voting-with-body-orientation hypothesis and are instead consistent with an emergent process for agreeing on travel direction, whereby group members respond to the orientation and trajectory of initiators as the group begins to move off (which here, we have broadly termed 'copying'). These findings differ from reports of buffalo and primates using body orientation to 'vote' on their preferred movement direction [11,14,18] and since the two mechanisms (voting, copying) can look similar [24,25], especially when observations are made at a coarser resolution than employed here, it is possible that some previous descriptions of voting in the literature [14,18] may actually represent copying.

While our results are consistent with shared decision-making mechanisms, support for this hypothesis could be interpreted as support for a general null hypothesis (symmetry of correlation

**Figure 4.** Goat decision-making mechanisms. Goats' (a) group speed (m s⁻¹); (b) decision parameter (where 0/1 = zero/perfect agreement among group members' orientation and departure direction), and (c) polar order parameter (where 0/1 = individuals point in random/the same directions) through time, with stationary, departing (moving), and travel phases indicated. Data are presented for n = 10 goats during n = 21 collective decisions with confidence intervals (95%; grey shaded area) obtained using 1000 bootstrap estimates. For a visualization of all data rather than confidence intervals see electronic supplementary material, figure S13. We assigned time zero (grey dashed line) as the time at which group mean heading was equal to the next destination. Group departure (first black dashed line) began at approximately −60 s and the moving phase took approximately 10 s before travelling phase began (second black dashed line). (d) average cross-correlation between speed and decision parameter; (e) the position of the peak (max time lag) and left–right symmetry (symmetry index) of the cross-correlation between speed and decision parameter (from (d); black points). Error bars (95% confidence intervals) were obtained using 1000 bootstrap estimates. (f) GPS and Accelerometer/magnetometer collared goats at the study site, photo credit: Lisa O'Bryan.

curve, and the time-shift at which correlation is maximized). Therefore, we employed three versions of an agent-based model—a 'copying' model, an 'all vote' model and a 'sub-group vote' model—formalizing our conceptual model (figure 1) and comparing model predictions with the empirical data. Our empirical data matched the predictions of the copying version of the agent-based model (figure 3a,b) and so we take this as evidence of goats responding to the orientation and trajectory of initiators as the group begins to move off, i.e. copying rules (cf. [12,33]), similar to that described for animal groups observed in the laboratory [9,22,23,44]. We were also reassured that the agent-based approach demonstrated that 'voting' is detectable using the decision parameter with our sample size (n = 21 events). Nevertheless, real-world data will be noisier than modelled data, so while the model supports the copying hypothesis, it is not conclusive proof. Furthermore, distinguishing between 'all-group' and 'sub-group' voting using our agent-based model was not possible (figure 3d,f) and so the model would need to be extended or refined to fully explore predictions of all-vote and sub-group-vote scenarios.

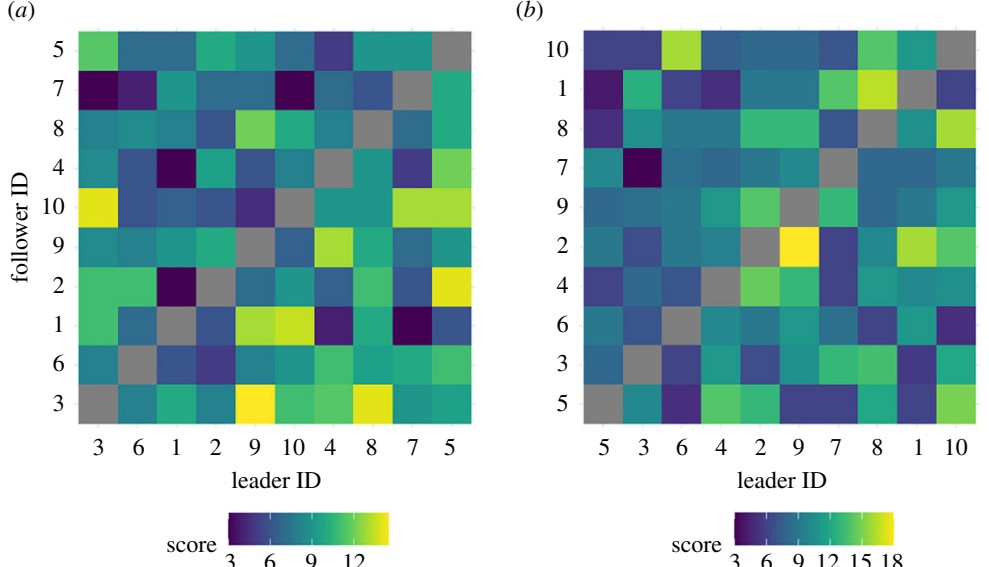

**Figure 5.** Leadership matrices. Leader–follower scores for pairs of goats ($n = 10$), 100 s during (*a*), and after (*b*), demarcated collective decisions ($n = 21$), ordered according to their leadership rank (normDS score; left to right, bottom to top), following [36]. Leadership scores were assigned at each time step (once per second), if the pair's magnetometer headings exceeded a correlation threshold of 0.9 (4 Hz). Cells are coloured with 'hot' (yellow) and 'cold' (blue) shades according to leadership scores. Data demonstrate no significant transitivity in either (*a*) departure phase ($t_{tri} = 0.13$, $p = 0.729$) or (*b*) travelling phase ($t_{tri} = -0.067$, $p = 0.618$).

Our agent-based model also provided insight into the likelihood of achieving travel consensus when copying or voting mechanism are employed and the important role of topological structure—i.e. how many neighbours each individual responded to (e.g. [45]). In the case of achieving travel consensus, 'voting' allowed modelled groups to come to consensus more often than 'non-voting' groups, which suggests that voting with body orientation before departure may increase democratic consensus in models (electronic supplementary material, figure S12). In the case of topological structure, varying the number of neighbours an individual responded to did not change the shape of the decision parameter curve, but was important for group cohesion (electronic supplementary material, figure S12). The model predicted the group would split more often when copying rules were employed among a few interacting neighbours (electronic supplementary material, figure S12). This information may be important for the further development of general models of collective decision-making in animal groups [3,46,47] and for researchers in swarm robotics (e.g. [48]) interested in understanding local (and optimal) information transfer [47].

We did not identify any consistency in the copying dynamics of goats from our analyses of pairwise correlations in goat heading. There was no significant transitivity among goat headings during (figure 5*a*) or after the departure phases (figure 5*b*), which would have indicated hierarchical copying dynamics [36]. However, we did find directional consistency within dyads [42,43], indicating that individual identity influences movement patterns locally [49]. Other research has found ungulate herd movements and leader–follower dynamics can be influenced by individual characteristics [50], and so it would be interesting to see if and how a more heterogeneous group—with individuals of different age classes, or sexes, for example—could result in greater transitivity and thus the potential for greater individual influence upon group decisions [28].

Based on our models and data, we have concluded that goats are probably responding to each other's orientation and movement during collective decisions. However, due to the generality of this 'copying' hypothesis, and its predictions being equivalent to that of a general null hypothesis (see discussion above), we must also consider the goats are simultaneously responding to a signal or cue that we did not capture [8,51–53]. For instance, vocal signals have been found to indicate preference for the timing [17,51] and direction [54] of collective movements, and we previously showed that goats may use vocalizations to facilitate the contraction of group spread in this system [30]. Therefore, while we did not observe any pattern in vocalizations during departure phases, we cannot rule this out, and although unlikely, it could also be that the goats are instantaneously responding to some environmental cue. Therefore, in future work, our model/methods could be integrated into studies that quantify more (and potentially all) of the animals' social and non-social stimuli (see [55]). Indeed, such work is needed to clarify the full range of social and environmental factors that may influence

group movements [12,49,55]. In addition, the insight of such studies could be increased by estimating which of these features the animals can actually sense at a given point in time or to what degree [56,57]. By combining knowledge of anatomy (e.g. eye location; frontal versus lateral) and perception (e.g. clarity of vision, sensitivity of hearing) with aerial mapping and fine-scale data on orientation (methods applied here) or behaviour [37] from animal-attached sensors [58], we may be able to achieve a full and complete picture of the mechanisms underlying individual and collective decision-making processes in free-ranging populations.

In sum, we have developed and tested a set of predictions that can be used to test mechanisms for making shared decisions regarding movement direction based on the orientations and trajectories of group members. Our approach also offers a route to begin to differentiate between 'simple' and 'complex' cognitive strategies used by individuals when making group decisions, since our findings suggest that simple rules of interaction can govern group movement decisions, without invoking cognitively complex abilities (i.e. vote counting and averaging) at the level of the individual.

Ethics. All applicable international, national and/or institutional guidelines for the care and use of animals were followed. Work was conducted under a Ministry of Environment and Tourism research permit (Research/ Collecting Permit 2009/2015) and was approved by the Rutgers Newark Animal Care and Use Committee.

Data accessibility. Data and relevant code for this research work are stored in GitHub: https://github.com/swarm-lab/ goatCollectiveDecision; https://github.com/sankeydan/voteABM and have been archived within the Zenodo repositories: https://doi.org/10.5281/zenodo.4265073 (goatCollectiveDecision); https://doi.org/10.5281/zenodo. 4246325 (voteABM).

Authors' contributions. D.W.E.S., I.F. and A.J.K. formulated hypotheses. L.R.O'B., A.J.K., P.H. and M.H. developed collars and hardware. G.C. and L.R.O'B. were responsible for fieldwork; L.R.O'B. performed data collection. D.W.E.S. processed the data; D.W.E.S. and S.G. analysed and visualized the data. D.W.E.S., L.R.O'B., S.G., G.C., I.F. and A.J.K. wrote the manuscript.

Competing interests. The authors declare that they have no conflict of interest.

Acknowledgements. We thank the Ministry of Lands and Resettlement and the Tsaobis beneficiaries, the Snyman and Wittreich families, the Gobabeb Research and Training Centre, and the Ministry of Environment and Tourism for fieldwork permission and support. We are grateful to the James S. McDonnell Foundation (grant no. 220020422) and Natural Environment Research Council (NE/H016600/3 and NE/M015351/1) for funding, the Tsaobis Baboon Project 2015 team and Herman Strydom for help, Lucy Hawkes, Rory Wilson and Liz Greenyer for discussion, and Layla King for support. We thank Dr Oliver Schülke (Associate Editor), Kevin Padian (Subject Editor), and two anonymous reviewers for improving the manuscript. This paper is a publication of the ZSL Institute of Zoology's Tsaobis Baboon Project.

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
