## [Peer Review File · Royal Society Open Science]

Review History

RSOS-201128.R0 (Original submission)

Review form: Reviewer 1

Is the manuscript scientifically sound in its present form?

Yes

Are the interpretations and conclusions justified by the results?

Yes

Is the language acceptable?

Yes

Do you have any ethical concerns with this paper?

No

Have you any concerns about statistical analyses in this paper?

No

Recommendation?

Accept with minor revision (please list in comments)

Comments to the Author(s)

This paper tests whether collective departures in goat groups show evidence of “voting” (align headings before speeding up) or “copying” (align headings and speed up at the same time) using magnetometer data and GPS data. The authors compare the time course of curves for a “decision parameter” (alignment with ultimate departure direction) and group speed, and use the maximum time lag and symmetry of the cross-correlation between these two values to distinguish hypotheses. They also develop an agent based model to explore whether the conceptual patterns can actually emerge from different rule sets, and explore the leader-follower dynamics within the herd using cross-correlations to see whether consistent leader-follower relationships exist during and after these events (they do) and whether a hierarchy emerges (it doesn't).

Overall, I enjoyed reading this paper. It is clearly presented and I think it makes a nice contribution by testing a hypothesis about “voting” by headings prior to departure that is often referenced but has not to my knowledge been quantitatively tested previously. The results about consistent leader-follower dynamics that are not hierarchical were also very interesting (and could benefit from further exploration, but perhaps in another paper).

A major issue of the paper in my view is that it draws its conclusions essentially from a null result: i.e. the fact that max time lag and asymmetry do not differ significantly from 0. The addition of the simulation model to show that the expected patterns can emerge does help with this to some extent, however as any real data is likely to be more noisy than simulated data this does not fully get around the issue. At the same time, I don't see much to be done about this issue, but I would suggest at least acknowledging it a bit more explicitly in the discussion (it is discussed in the current MS, but in a way that indicates the agent-based model removes this concern, whereas in my view having the agent-based model helps but does not completely remove the concern because of the reason mentioned above).

A further issue (related to the above) is that even if the null result is accepted and hence the paper can rule out voting (or sub-voting), the paper does not in the end support a particular mechanism. The use of the term “copying” in my view kind of hides this issue by making it sound as if the mechanism is really understood when in fact there are many possible mechanisms one could imagine that would cause the simultaneous rise of the decision parameter and the speed. I don't think it's necessary to completely figure out the mechanism of decision-making in order to make this a valid piece of work and a useful contribution to the literature, however I would suggest to make this point a bit more clearly in the manuscript (the discussion which talks about the need to quantify the influence of social and non-social stimuli could be a good place to integrate this point about the vagueness / generality of the “copying” mechanism, and I would also make this point earlier on to avoid misleading readers about how non-specific the “copying mechanism” really is).

Line edits:

L35-36: Sentence fragment

L69: “strength of preferences” - this is a bit confusingly worded because it could be interpreted as an individual's strength of preference, whereas what is meant is the support across all group members for an options - suggest rewording

L79: This might be a good point to start to address my second point above. It's not necessarily "copying" of movements and orientations, it could be simple attraction to conspecifics, for instance.

L139-140: I think this is a mistake in the writing, as the later part shows why consistency and hierarchical dynamics are not the same thing

Figure 1 / Methods: Perhaps it is just me, but I am having a hard time understanding conceptually why the symmetry index predictions are as they are. This would be worth explaining in a bit more detail.

Figure 1 (and a few others): some of the figures do not come out well in black and white. I would suggest changing some of the color schemes to (or adding additional non-colour indicators e.g. dotted lines) to help with this.

L321-322: As these are different species, I wouldn't say the results are "at odds" - however the results do suggest that it would be worth re-examining the hypothesised voting mechanism in these other species using these methods.

L357-358: sentence fragment

L394: I see what you mean, but these are not technically "quantitative predictions" - even though they use quantitative methods the predictions are ultimately qualitative. Suggest rewording.

Figure 4: Given that there are only 21 lines to show, could you just show all of them (in lighter color) rather than the confidence intervals? This would give a better sense of how the raw data looks, and potentially differences in the time course (which the confidence intervals somewhat hide).

Supplemental video 1: I couldn't get the video to completely work on my machine. Quicktime doesn't work at all and VLC video works but no sound. This was unfortunate as at least from the images this video abstract looked helpful!

Review form: Reviewer 2

Is the manuscript scientifically sound in its present form?

No

Are the interpretations and conclusions justified by the results?

No

Is the language acceptable?

Yes

Do you have any ethical concerns with this paper?

No

Have you any concerns about statistical analyses in this paper?

No

Recommendation?

Reject

Comments to the Author(s)

General comment:

In this manuscript, the authors aimed at differentiating between voting and copying mechanisms. This question is indeed crucial. However, I have several concerns about this study: the first is about the few numbers of events that have been analyzed and, secondly, the authors' lack of hindsight in their interpretations. Moreover, the interpretation/analysis of the literature was biased in order to make their work totally ground-breaking. As much I found their first paper on goat's vocalizations very good, as much this one is disappointing, despite its interesting purpose.

Specific comments:

Introduction

When stating their hypotheses, the authors introduced a temporal distinction between voting and copying mechanisms that has not been exposed early when they made reference to the literature. That is not logical as this is at the basis of the article's reasoning (cf lines 123 to 131). This temporal difference must be reported line 53 and then line 72 when they described previous studies.

Line 72: repetition. I advise to remove the end of the sentence "and in the absence of established procedures such as voting"

Line 100-101: it has been possible to collect this information in semi-free ranging condition on several species. Please rewrite the sentence by stating these previous studies in capuchin monkeys, Tonkean macaques and horses (Meunier et al. 2008; Sueur et al., 2010; Briard et al, 2017; Gérard, Valençon et al., 2020).

Line 102 to 104: I perfectly understand that technology renders such study feasible, but it is not a goal in itself. Disentangling between voting or copying mechanisms is the main question of this study and must not be replaced by a technical aim. This part needs to be moved to the methods section.

Line 107-110: again, the authors evoked at first a technical reason for studying goats whereas this model is a perfect one for investigating the question of voting or copying mechanisms. I advise to start this paragraph by the sentences between lines 109 and 117. The authors could then add the technical justification and then proceed with their assumptions.

Methods

The method for obtaining speed and heading is very clearly stated and easy to understand.

Line 176: The duration of these 23 pre-departure periods is not clearly stated. After further reading (especially S4), I understood it corresponds to one hundred seconds which is very short. Can the authors give arguments for this choice?

Line 201: there is a useless dot.

Line 221: I don't understand at that point the usefulness of the polar order. This needs a more detailed explanation/justification.

Results

Line 291, the authors reported that about 50-60 seconds before departure, the group become more aligned with the departure direction, identified by a steep rise in decision parameter. Given their assumptions - goats either: (1) used body orientation to vote on their preferred direction of travel prior to departure (voting hypothesis) or (2) simultaneously matched their orientation and trajectory with neighbors as the group moves off (copying hypothesis) - how can the authors reject their voting hypothesis? I agree that figure 4 and S9 are convincing but apparently contradictory to the sentence I reported above.

Discussion

Line 327: I agree that results obtained by Prins and Kummer were based on direct observations. On the contrary, in their study, Sueur and colleagues have continuously filmed the behavior of their groups and have been able (thanks to videos) to distinguish between pre-departure

behaviors (orientations and others) and post-departure behaviors. As a consequence, the authors cannot reject the conclusions of Sueur et al's study as they did.

Line 344: The authors stated that: "given their data fit the 'copying model', no further work in this direction is necessary for the conclusions of our empirical study". I'm sorry but 10 days of data collection leading to "only" 21 decisions are not a huge set of data. The authors need to be more cautious in their conclusions even if they obtained many measurements for each event and built agent-based models. Moreover, this conclusion is in opposite to what they develop lines 352-358. Lines 360-370: Goats have individual characteristics that might explain directional consistency within dyads. Again, more data would probably have helped the authors to explain leader-follower dynamics. In the same line, in S6, the authors reported that some information was missing.

In the end, I consider the discussion to be confusing.

Decision letter (RSOS-201128.R0)

Dear Dr Sankey

The Editors assigned to your paper RSOS-201128 "Consensus of travel direction achieved by simple copying, not voting, in free-ranging goats" have now received comments from reviewers and would like you to revise the paper in accordance with the reviewer comments and any comments from the Editors. Please note this decision does not guarantee eventual acceptance.

Please submit your revised manuscript and required files (see below) no later than 21 days from today's (ie 20-Aug-2020) date. Note: the ScholarOne system will 'lock' if submission of the revision is attempted 21 or more days after the deadline. If you do not think you will be able to meet this deadline please contact the editorial office immediately.

on behalf of Dr Oliver Schülke (Associate Editor) and Kevin Padian (Subject Editor)
openscience@royalsociety.org

Subject Editor Comments to Author (Professor Kevin Padian):

Comments to the Author:

I agree with the AE's recommendation. Best wishes with your revisions, and if you need more time, please contact the editorial office.

Associate Editor Comments to Author (Dr Oliver Schülke):

Comments to the Author:

Dear Dr. Sankey,
as the associate editor handling your submission, I have received comments from two expert referees. based on their comments and my own reading of the manuscript I will suggest to my editor to as for a major revision of your work before reconsidering it. Except for an underappreciated sample size issue both reviewers are happy with the data, analyses, and model. Yet, both raise issues concerning the writing with respect to the uniqueness of this work and the certainty with which results are presented. We all agree that new technology allows us to assess collective behavior with unprecedented detail. This does not make previous efforts to glean the information from videos and the results of those studies obsolete. I encourage you to very carefully change your text in accordance with the comments and look forward to receiving a new version of your work at your earliest convenience.

With kind regards,
Oliver Schülke
(Associate Editor RSOS)

Reviewer comments to Author:

Reviewer: 1

Comments to the Author(s)

This paper tests whether collective departures in goat groups show evidence of “voting” (align headings before speeding up) or “copying” (align headings and speed up at the same time) using magnetometer data and GPS data. The authors compare the time course of curves for a “decision parameter” (alignment with ultimate departure direction) and group speed, and use the maximum time lag and symmetry of the cross-correlation between these two values to distinguish hypotheses. They also develop an agent based model to explore whether the conceptual patterns can actually emerge from different rule sets, and explore the leader-follower dynamics within the herd using cross-correlations to see whether consistent leader-follower relationships exist during and after these events (they do) and whether a hierarchy emerges (it doesn't).

Overall, I enjoyed reading this paper. It is clearly presented and I think it makes a nice contribution by testing a hypothesis about “voting” by headings prior to departure that is often referenced but has not to my knowledge been quantitatively tested previously. The results about consistent leader-follower dynamics that are not hierarchical were also very interesting (and could benefit from further exploration, but perhaps in another paper).

A major issue of the paper in my view is that it draws its conclusions essentially from a null result: i.e. the fact that max time lag and asymmetry do not differ significantly from 0. The addition of the simulation model to show that the expected patterns can emerge does help with this to some extent, however as any real data is likely to be more noisy than simulated data this does not fully get around the issue. At the same time, I don't see much to be done about this issue, but I would suggest at least acknowledging it a bit more explicitly in the discussion (it is discussed in the current MS, but in a way that indicates the agent-based model removes this concern, whereas in my view having the agent-based model helps but does not completely remove the concern because of the reason mentioned above).

A further issue (related to the above) is that even if the null result is accepted and hence the paper can rule out voting (or sub-voting), the paper does not in the end support a particular mechanism. The use of the term “copying” in my view kind of hides this issue by making it sound as if the mechanism is really understood when in fact there are many possible mechanisms one could imagine that would cause the simultaneous rise of the decision parameter and the speed. I don't think it's necessary to completely figure out the mechanism of decision-making in order to make this a valid piece of work and a useful contribution to the literature, however I would suggest to make this point a bit more clearly in the manuscript (the discussion which talks about the need to quantify the influence of social and non-social stimuli could be a good place to integrate this point about the vagueness / generality of the “copying” mechanism, and I would also make this point earlier on to avoid misleading readers about how non-specific the “copying mechanism” really is).

Line edits:

L35-36: Sentence fragment

L69: “strength of preferences” - this is a bit confusingly worded because it could be interpreted as an individual's strength of preference, whereas what is meant is the support across all group members for an options - suggest rewording

L79: This might be a good point to start to address my second point above. It's not necessarily “copying” of movements and orientations, it could be simple attraction to conspecifics, for instance.

L139-140: I think this is a mistake in the writing, as the later part shows why consistency and hierarchical dynamics are not the same thing

Figure 1 / Methods: Perhaps it is just me, but I am having a hard time understanding conceptually why the symmetry index predictions are as they are. This would be worth explaining in a bit more detail.

Figure 1 (and a few others): some of the figures do not come out well in black and white. I would suggest changing some of the color schemes to (or adding additional non-colour indicators e.g. dotted lines) to help with this.

L321-322: As these are different species, I wouldn't say the results are "at odds" - however the results do suggest that it would be worth re-examining the hypothesised voting mechanism in these other species using these methods.

L357-358: sentence fragment

L394: I see what you mean, but these are not technically "quantitative predictions" - even though they use quantitative methods the predictions are ultimately qualitative. Suggest rewording.

Figure 4: Given that there are only 21 lines to show, could you just show all of them (in lighter color) rather than the confidence intervals? This would give a better sense of how the raw data looks, and potentially differences in the time course (which the confidence intervals somewhat hide).

Supplemental video 1: I couldn't get the video to completely work on my machine. Quicktime doesn't work at all and VLC video works but no sound. This was unfortunate as at least from the images this video abstract looked helpful!

Reviewer: 2

Comments to the Author(s)

General comment:

In this manuscript, the authors aimed at differentiating between voting and copying mechanisms. This question is indeed crucial. However, I have several concerns about this study: the first is about the few numbers of events that have been analyzed and, secondly, the authors' lack of hindsight in their interpretations. Moreover, the interpretation/analysis of the literature was biased in order to make their work totally ground-breaking. As much I found their first paper on goat's vocalizations very good, as much this one is disappointing, despite its interesting purpose.

Specific comments:

Introduction

When stating their hypotheses, the authors introduced a temporal distinction between voting and copying mechanisms that has not been exposed early when they made reference to the literature. That is not logical as this is at the basis of the article's reasoning (cf lines 123 to 131). This temporal difference must be reported line 53 and then line 72 when they described previous studies.

Line 72: repetition. I advise to remove the end of the sentence "and in the absence of established procedures such as voting"

Line 100-101: it has been possible to collect this information in semi-free ranging condition on several species. Please rewrite the sentence by stating these previous studies in capuchin monkeys, Tonkean macaques and horses (Meunier et al. 2008; Sueur et al., 2010; Briard et al, 2017; Gérard, Valençon et al., 2020).

Line 102 to 104: I perfectly understand that technology renders such study feasible, but it is not a goal in itself. Disentangling between voting or copying mechanisms is the main question of this study and must not be replaced by a technical aim. This part needs to be moved to the methods section.

Line 107-110: again, the authors evoked at first a technical reason for studying goats whereas this model is a perfect one for investigating the question of voting or copying mechanisms. I advise to start this paragraph by the sentences between lines 109 and 117. The authors could then add the technical justification and then proceed with their assumptions.

Methods

The method for obtaining speed and heading is very clearly stated and easy to understand.

Line 176: The duration of these 23 pre-departure periods is not clearly stated. After further reading (especially S4), I understood it corresponds to one hundred seconds which is very short. Can the authors give arguments for this choice?

Line 201: there is a useless dot.

Line 221: I don't understand at that point the usefulness of the polar order. This needs a more detailed explanation/justification.

Results

Line 291, the authors reported that about 50-60 seconds before departure, the group become more aligned with the departure direction, identified by a steep rise in decision parameter. Given their assumptions - goats either: (1) used body orientation to vote on their preferred direction of travel prior to departure (voting hypothesis) or (2) simultaneously matched their orientation and trajectory with neighbors as the group moves off (copying hypothesis) - how can the authors reject their voting hypothesis? I agree that figure 4 and S9 are convincing but apparently contradictory to the sentence I reported above.

Discussion

Line 327: I agree that results obtained by Prins and Kummer were based on direct observations. On the contrary, in their study, Sueur and colleagues have continuously filmed the behavior of their groups and have been able (thanks to videos) to distinguish between pre-departure behaviors (orientations and others) and post-departure behaviors. As a consequence, the authors cannot reject the conclusions of Sueur et al's study as they did.

Line 344: The authors stated that: "given their data fit the 'copying model', no further work in this direction is necessary for the conclusions of our empirical study". I'm sorry but 10 days of data collection leading to "only" 21 decisions are not a huge set of data. The authors need to be more cautious in their conclusions even if they obtained many measurements for each event and built agent-based models. Moreover, this conclusion is in opposite to what they develop lines 352-358.

Lines 360-370: Goats have individual characteristics that might explain directional consistency within dyads. Again, more data would probably have helped the authors to explain leader-follower dynamics. In the same line, in S6, the authors reported that some information was missing.

In the end, I consider the discussion to be confusing.

===PREPARING YOUR MANUSCRIPT===

===PREPARING YOUR REVISION IN SCHOLARONE===

Author's Response to Decision Letter for (RSOS-201128.R0)

See Appendix A.

Decision letter (RSOS-201128.R1)

Dear Dr Sankey,

It is a pleasure to accept your manuscript entitled "Consensus of travel direction achieved by simple copying, not voting, in free-ranging goats" in its current form for publication in Royal Society Open Science.

At this stage, we ask that you please archive your GitHub code within the Zenodo repository: <https://guides.github.com/activities/citable-code/>. By doing this, a formal, citable DOI will be associated with your data record, and an open license (CC-BY preferred) can be applied to your data. We would then ask that you please update your data availability statement to read as:

"Data and relevant code for this research work are stored in GitHub: [GitHub URL here] and have been archived within the Zenodo repository: <https://doi.org/zenodo.....> [ref number].

on behalf of Dr Oliver Schülke (Associate Editor) and Kevin Padian (Subject Editor)
openscience@royalsociety.org

Appendix A

Associate Editor Comments to Author (Dr Oliver Schülke):

Comments to the Author:

Dear Dr. Sankey,
as the associate editor handling your submission, I have received comments from two expert referees. based on their comments and my own reading of the manuscript I will suggest to my editor to ask for a major revision of your work before reconsidering it. Except for an underappreciated sample size issue both reviewers are happy with the data, analyses, and model. Yet, both raise issues concerning the writing with respect to the uniqueness of this work and the certainty with which results are presented. We all agree that new technology allows us to assess collective behavior with unprecedented detail. This does not make previous efforts to glean the information from videos and the results of those studies obsolete. I encourage you to very carefully change your text in accordance with the comments and look forward to receiving a new version of your work at your earliest convenience.

With kind regards,
Oliver Schülke
(Associate Editor RSOS)

We thank the Associate Editor for their comments. We have addressed the concerns of the two Reviewers, and provide replies in bold text below. In our revision we have also been careful not to be dismissive of direct observation of animal groups, or suggest that direct observations are obsolete. This was not our intention at all in our original submission; indeed – all of the authors of this manuscript rely heavily upon direct observations in their research.

Reviewer comments to Author:

Reviewer: 1

Comments to the Author(s)

This paper tests whether collective departures in goat groups show evidence of “voting” (align headings before speeding up) or “copying (align headings and speed up at the same time) using magnetometer data and GPS data. The authors compare the time course of curves for a “decision parameter” (alignment with ultimate departure direction) and group speed, and use the maximum time lag and symmetry of the cross-correlation between these two values to distinguish hypotheses. They also develop an agent based model to explore whether the conceptual patterns can actually emerge from different rule sets, and explore the leader-follower dynamics within the herd using cross-correlations to see whether consistent leader-follower relationships exist during and after these events (they do) and whether a hierarchy emerges (it doesn't).

Overall, I enjoyed reading this paper. It is clearly presented and I think it makes a nice contribution by testing a hypothesis about “voting” by headings prior to departure that is often referenced but has not to my knowledge been quantitatively tested previously. The results about consistent leader-follower dynamics that are not hierarchical were also very interesting (and could benefit from further exploration, but perhaps in another paper).

A major issue of the paper in my view is that it draws its conclusions essentially from a null result: i.e. the fact that max time lag and asymmetry do not differ significantly from 0. The addition of the simulation model to show that the expected patterns can emerge does help with this to some extent, however as any real data is likely to be more noisy than simulated data this does not fully get around the issue. At the same time, I don't see much to be done about this issue, but I would suggest at least acknowledging it a bit more explicitly in the discussion (it is discussed in the current MS, but in a way that indicates the agent-based

model removes this concern, whereas in my view having the agent-based model helps but does not completely remove the concern because of the reason mentioned above).

We thank the Reviewer for the excellent comments, and are very pleased by their assessment. In response to the Reviewer concerns, we now explicitly acknowledge that our copying hypothesis does not differ from a null hypothesis, and devote a paragraph to discussing this point in the context of the agent-based model we present (Lines 316-332) and then later in the discussion have a paragraph where we discuss other mechanisms that would fit our data and how to investigate these in future work (Lines 361-384).

A further issue (related to the above) is that even if the null result is accepted and hence the paper can rule out voting (or sub-voting), the paper does not in the end support a particular mechanism. The use of the term “copying” in my view kind of hides this issue by making it sound as if the mechanism is really understood when in fact there are many possible mechanisms one could imagine that would cause the simultaneous rise of the decision parameter and the speed. I don't think it's necessary to completely figure out the mechanism of decision-making in order to make this a valid piece of work and a useful contribution to the literature, however I would suggest to make this point a bit more clearly in the manuscript (the discussion which talks about the need to quantify the influence of social and non-social stimuli could be a good place to integrate this point about the vagueness / generality of the “copying” mechanism, and I would also make this point earlier on to avoid misleading readers about how non-specific the “copying mechanism” really is).

We agree with the Reviewer on this point. Where we first introduce the “copying mechanism” we now state in the introduction that shared decisions can be achieved by “...individuals copying each other's movements and orientations (e.g. attraction to each other's motion)...” (Line 76). This makes it clearer at the outset that we are talking about individuals responding to one another's positions and motion (which could be achieved by a variety of behavioural rules/mechanisms). We have also extended our discussion on the vagueness / generality of the “copying” mechanism in the discussion as suggested, stating “...consistent with an emergent process for agreeing on travel direction, whereby group members respond to the orientation and trajectory of initiators as the group begins to move off (which here, we have broadly termed “copying”).” (Lines 305-309), and “...due to the generality of this “copying” hypothesis, and its predictions being equivalent to that of a general null hypothesis (see discussion above) we must also consider the goats are simultaneously responding to a signal or cue that we did not capture (Fossey 1972; Black 1988; Boinski 1993; Sueur and Petit 2010).” (Lines 362-366).

Line edits:

L35-36: Sentence fragment

Now changed to “Our findings highlight the role of simple behavioural rules for collective decision making by animal groups.” (Line 33-34)

L69: “strength of preferences” - this is a bit confusingly worded because it could be interpreted as an individual's strength of preference, whereas what is meant is the support across all group members for an options - suggest rewording

We have now changed this sentence to “structured decision making whereby each group member is able to assess the relative support for different options among their group-mates” (Line 66-68).

L79: This might be a good point to start to address my second point above. It's not necessarily "copying" of movements and orientations, it could be simple attraction to conspecifics, for instance.

We consider attraction to neighbours motion as 'copying' in this context (since they all begin moving and everyone moves) and so have emphasised this: "*Under this scenario, individuals copying each other's movement (e.g. attraction to each other's positions and/or orientations) results in a consensus across individuals' directional preferences in both real and simulated animal groups (Couzin et al. 2005; Conradt et al. 2009) (Line 75-79).*

L139-140: I think this is a mistake in the writing, as the later part shows why consistency and hierarchical dynamics are not the same thing

Now changed (Line 126-129)

Figure 1 / Methods: Perhaps it is just me, but I am having a hard time understanding conceptually why the symmetry index predictions are as they are. This would be worth explaining in a bit more detail.

Symmetry index is a statistic for distinguishing between hypotheses using the cross-correlation curve. It is calculated by subtracting the area under the curve to the left of zero by the area under the curve to the right of zero. A negative symmetry index indicates that navigational decisions are made before speed increases. We have now explained this in greater detail in the Figure 1 legend as suggested by the Reviewer.

Figure 1 (and a few others): some of the figures do not come out well in black and white. I would suggest changing some of the color schemes to (or adding additional non-colour indicators e.g. dotted lines) to help with this.

We have now changed Figure 1 so that the colour palette is colour-blind and grey-scale friendly (changing colours and using dashed lines).

L321-322: As these are different species, I wouldn't say the results are "at odds" - however the results do suggest that it would be worth re-examining the hypothesised voting mechanism in these other species using these methods.

We agree and have removed this sentence.

L357-358: sentence fragment

Now changed (Lines 344-347)

L394: I see what you mean, but these are not technically "quantitative predictions" - even though they use quantitative methods the predictions are ultimately qualitative. Suggest rewording.

We agree and have removed "quantitative" from the sentence (Line 386).

Figure 4: Given that there are only 21 lines to show, could you just show all of them (in lighter color) rather than the confidence intervals? This would give a better sense of how the raw data looks, and potentially differences in the time course (which the confidence intervals somewhat hide).

We revised the figure to show all the lines in lighter colour and the main effect in bold, but because of the variation observed it does make the panel look untidy, and so we included this plot as a supplemental figure (Figure S13).

Supplemental video 1: I couldn't get the video to completely work on my machine. Quicktime doesn't work at all and VLC video works but no sound. This was unfortunate as at least from the images this video abstract looked helpful!

There is no sound with the video abstract.

Reviewer: 2

Comments to the Author(s)

General comment:

In this manuscript, the authors aimed at differentiating between voting and copying mechanisms. This question is indeed crucial. However, I have several concerns about this study: the first is about the few numbers of events that have been analyzed and, secondly, the authors' lack of hindsight in their interpretations. Moreover, the interpretation/analysis of the literature was biased in order to make their work totally ground-breaking. As much I found their first paper on goat's vocalizations very good, as much this one is disappointing, despite its interesting purpose.

Specific comments:

Introduction

When stating their hypotheses, the authors introduced a temporal distinction between voting and copying mechanisms that has not been exposed early when they made reference to the literature. That is not logical as this is at the basis of the article's reasoning (cf lines 123 to 131). This temporal difference must be reported line 53 and then line 72 when they described previous studies.

We agree that the order in which we introduce concepts was not ideal. We have therefore re-ordered the introductory text as follows: The first paragraph (Lines 41-49) introduces the general background on group decision-making. The second paragraph (Lines 51-68) and third paragraph (Lines 70-81) introduce the specific background on voting, and copying, respectively. Then, paragraph four (Lines 83-95) discusses why differentiating between these two mechanisms is challenging. Then, the last two paragraphs (five and paragraph) introduce the system and the specific details of our hypothesis and predictions (Lines 97-129). We now also explicitly state the temporal difference associated with the two competing hypotheses (Lines 52-53 and Lines 76-77).

Line 72: repetition. I advise to remove the end of the sentence "and in the absence of established procedures such as voting"

Now removed (Line 70-71)

Line 100-101: it has been possible to collect this information in semi-free ranging condition on several species. Please rewrite the sentence by stating these previous studies in capuchin monkeys, Tonkean macaques and horses (Meunier et al. 2008; Sueur et al., 2010; Briard et al, 2017; Gérard, Valençon et al., 2020).

We agree this was poorly worded with reference to what has been achieved in previous work. Rather than directly compare to direct observation work, we reword: "...to distinguish between voting and copying mechanisms in the wild requires that

researchers continuously and simultaneously measure group members' orientation and movement towards a target destination, and this is logistically challenging (Strandburg-Peshkin et al. 2015; Bonnell et al. 2016; Flack et al. 2018; King et al. 2018)." (Lines 85-88)

Line 102 to 104: I perfectly understand that technology renders such study feasible, but it is not a goal in itself. Disentangling between voting or copying mechanisms is the main question of this study and must not be replaced by a technical aim. This part needs to be moved to the methods section.

We agree, and have re-worked the final paragraphs of the introduction and removed all the methods description and reference to the supplemental material that the Referee mentions here.

Line 107-110: again, the authors evoked at first a technical reason for studying goats whereas this model is a perfect one for investigating the question of voting or copying mechanisms. I advise to start this paragraph by the sentences between lines 109 and 117. The authors could then add the technical justification and then proceed with their assumptions.

We hope our reworking of the final paragraphs of the introduction and further changes in the methods and supplemental material (see responses above) satisfy this point.

Methods

The method for obtaining speed and heading is very clearly stated and easy to understand. Line 176: The duration of these 23 pre-departure periods is not clearly stated. After further reading (especially S4), I understood it corresponds to one hundred seconds which is very short. Can the authors give arguments for this choice?

The pre-departure (stationary) periods lasted different lengths of time and we present data on the 100 seconds before departure (e.g. Figure 4a). We now explain this in more detail in the Supplementary Methods (*Termination of previous movement*; Page 10). If we extended this time-window beyond 100s we would start seeing changes in motion and heading relating to movements for some events, which by definition do not represent pre-departure periods, defined as "the stationary period in a new resting or foraging zone" (Petit O, Bon R. Decision-making processes: the case of collective movements. *Behav Processes*. 2010;84(3):635–47.) We have now checked and edited the definition and exploration of these time-windows in the Supplemental Material S4.

Line 201: there is a useless dot.

Now removed.

Line 221: I don't understand at that point the usefulness of the polar order. This needs a more detailed explanation/justification.

The polar order parameter is required to confirm group members aligned with each other during departure, and not only to the travel destination. It is very unlikely they could do anything else, but given the use of the parameter in collective behaviour research, we think it is a useful addition. In the methods we now state: "*To check goat alignment regardless of the group mean orientation and decision parameter, we also calculated a polar order (or, alignment) parameter using the 'rho.circular' function (R package: CircStats; Lund & Agostinelli 2007). This parameter is required to confirm*

group members aligned with each other during departure (and not only to the travel destination)." Lines 208-215.

Results

Line 291, the authors reported that about 50-60 seconds before departure, the group become more aligned with the departure direction, identified by a steep rise in decision parameter. Given their assumptions - goats either: (1) used body orientation to vote on their preferred direction of travel prior to departure (voting hypothesis) or (2) simultaneously matched their orientation and trajectory with neighbors as the group moves off (copying hypothesis) - how can the authors reject their voting hypothesis? I agree that figure 4 and S9 are convincing but apparently contradictory to the sentence I reported above.

We thank the Reviewer for highlighting this. The original text wrongly used "departure" when it should have read "movement phase". We have now changed the text to read "*During the movement phase (Figure 4a) the group become more aligned with the departure direction, identified by a steep rise in decision parameter.*" (Lines 278-280).

Discussion

Line 327: I agree that results obtained by Prins and Kummer were based on direct observations. On the contrary, in their study, Sueur and colleagues have continuously filmed the behavior of their groups and have been able (thanks to videos) to distinguish between pre-departure behaviors (orientations and others) and post-departure behaviors. As a consequence, the authors cannot reject the conclusions of Sueur et al's study as they did.

This is an excellent point the Reviewer makes – we should have noticed this error; our apologies. We have now edited the text and no longer cite Sueur et al. where we make this statement and have "toned" down the sentence, and now state: "*These findings differ to reports of buffalo and primates using body orientation to "vote" on their preferred movement direction (Kummer 1995; Prins 1996; Sueur et al. 2010) and since the two mechanisms (voting, copying) can look similar (Couzin and King 2010; King and Sueur 2011), especially when observations are made at a coarser resolution than employed here, it is possible that some previous descriptions of voting in the literature (Kummer 1995; Prins 1996) may actually represent copying.*" (Lines 310-315).

Line 344: The authors stated that: "given their data fit the 'copying model', no further work in this direction is necessary for the conclusions of our empirical study". I'm sorry but 10 days of data collection leading to "only" 21 decisions are not a huge set of data. The authors need to be more cautious in their conclusions even if they obtained many measurements for each event and built agent-based models. Moreover, this conclusion is in opposite to what they develop lines 352-358.

Our sentence was poorly worded, and was not actually needed. What we were trying to explain (poorly) was that our agent-based model was not able to distinguish between voting and sub-group voting well (but did match conceptual and empirical data for the copying hypothesis). We have therefore re-written this section and now state: "*However, distinguishing between "all-group" and "sub-group" voting using our agent-based model was not possible (Figure 3d,f) and so the model would need to be extended or refined to fully explore predictions of all-vote and sub-group-vote scenarios.*" (Lines 329-333).

Lines 360-370: Goats have individual characteristics that might explain directional consistency within dyads. Again, more data would probably have helped the authors to

explain leader-follower dynamics. In the same line, in S6, the authors reported that some information was missing.

We agree – and suggest future work should test this properly (Lines 356-360). Note that our statement in Figure S6 that said “...for which we had full and complete information” sounded like information was missing for these goats, which it is not (it is complete). It was not worded well and we have deleted this statement.

In the end, I consider the discussion to be confusing.

We have made several major changes to the content and order of the discussion (see tracked changes version of manuscript) which has hopefully improved accuracy and clarity.